# Validity and Reliability of the COVID-19 Knowledge, Attitude and Behavior Scale

**DOI:** 10.3390/vaccines11020317

**Published:** 2023-01-31

**Authors:** Serol Deveci, Celalettin Cevik, Hakan Baydur, Fatih Onsuz, Selma Tosun, Alp Ergor

**Affiliations:** 1Manisa Şehzadeler District Health Directorate, Manisa Provincial Health Directorate, 45140 Manisa, Türkiye; 2Department of Public Health Nursing, Faculty of Health Sciences, Balikesir University, 10145 Balikesir, Türkiye; 3Department of Social Work, Faculty of Health Sciences, Manisa Celal Bayar University, 45140 Manisa, Türkiye; 4Department of Public Health, Faculty of Medicine, Eskişehir Osmangazi University, 26040 Eskişehir, Türkiye; 5İzmir Bozyaka Training and Research Hospital Infectious Diseases and Clinical Microbiology Clinic, 35220 İzmir, Türkiye; 6Department of Public Health, Faculty of Medicine, Dokuz Eylul University, 35220 Konak, Türkiye

**Keywords:** COVID-19, knowledge, attitude, practice, scale development

## Abstract

Background: The aim of this research is to develop a scale that will evaluate the knowledge, attitudes and behaviors of employees about COVID-19 and to test its validity and reliability. Methodology: The methodological type of research was used between August–November 2020, under observation in organized industrial zones. Information was collected from a total of 543 employees. Confirmatory factor analysis and correlation analysis were performed for the value, item–total correlations and construct validity. SPSS 25.0 (IBM Inc., Armonk, NY, USA), Jasp 0.14 (University of Amsterdam) and Lisrel 9.1(Scientific Software International, Inc., Chapel Hill, NC, USA) programs were used in the analysis. Results: 83.1% of the participants in the study are male, the average age is 37.4 ± 8.0, 76.1% are married, and 49.4% are high school graduates. The Cronbach alpha value of the COVID-19 information part is 0.86 in total, the contamination information dimension is 0.71 and the protection information dimension is 0.84. The COVID-19 attitude section consists of four sub-dimensions and 13 items classified within the framework of the health belief model. In summary, the goodness of fit values for the knowledge, attitude and behavior sections, respectively, are: RMSEA values 0.05, 0.03 and 0.04; CFI values 0.98, 0.98 and 0.99; GFI values 0.97, 0.97 and 0.99. Conclusions: It has been determined that the internal consistency of the COVID-19 knowledge, attitude and behavior scale conducted on employees is high and compatible, and its validity findings are sufficient. The scale is recommended as an applicable tool to measure COVID-19 knowledge, attitudes and behaviors.

## 1. Introduction

The COVID-19 pandemic, which emerged at the end of 2019, is an important public health problem that negatively affects the socioeconomic, political and social structure. According to the data of the World Health Organization (WHO), as of the middle of August 2022, 6.425.422 deaths had occurred and 218 countries had been affected by COVID-19 [1]. COVID-19 is a contagious disease affecting health, economy and society that can cause serious symptoms such as fever, dry cough, difficulty in breathing, chest pain, and difficulty in speaking and moving [1].

SARS-CoV-2 can be transmitted from person to person through droplets and indirect contact with contaminated objects [2]. Although attempts are being made to bring the SARS-CoV-2 pandemic under control, there have been changes in dietary habits, physical activity levels, consumer behaviors, education-teaching methods and daily life. Given the spread of SARS-CoV-2 and its impact on human health, the WHO has recommended strategies to control this pandemic, such as traffic restriction, cancellation of social gatherings, community engagement, home quarantine, and case and contact tracing [3]. Despite all these strategies, reasons such as the limited treatments available during the COVID-19 pandemic, the fact that community immunity will take time with the vaccine, and the uncertainties of infectious disease situations which cause panic make it more important to determine the knowledge, attitudes and behaviors of people [4]. In the limited number of studies evaluating knowledge, attitudes and behaviors related to COVID-19, knowledge, attitude and behavior were evaluated using various questions. In these studies, it is seen that the knowledge score is sufficient and good [5,6,7], the attitude score is inconstant [5,7] and the behavioral score is low at the community level, where it is mostly high on health workers [5,6,7,8]. Looking at the literature, when looking at the factors that determine the level of COVID-19 knowledge, attitude and behavior, different predictors for the PDP of COVID-19, including socio-demographic characteristics (age, gender, economic status, race, marital status, occupation and language), were found [9,10,11,12]. According to research conducted in Vietnam, Pakistan, China, Saudi Arabia and Malaysia, occupation was found to be a determinant of knowledge and attitude [9,12,13].

Until now, in addition to the use of personal protective equipment, the identification of patients with symptoms and the separation of cases and contacts from healthy ones have been achieved. However, the appropriate knowledge, attitudes and behaviors of individuals can also increase the correct implementation of COVID-19 prevention measures. This can be achieved through knowing the knowledge, attitude and application awareness of individuals towards COVID-19. On the other hand, since there is no developed scale measuring the knowledge, attitude and behavior of individuals regarding COVID-19 [2,14], it was decided to conduct this study. The basic components that are essential for individuals to stay healthy can be evaluated within the scope of knowledge, attitudes and behaviors. In other words, the foundations of staying healthy are related to what the individual knows and believes, and how she/he acts. Thus, the individual can protect and maintain his/her health. The most important condition for evaluating this objectively is to create concrete measurement tools and to monitor the results based on self-reporting.

COVID-19 knowledge, attitude and behavior scale development and analysis includes the following steps. There are three basic elements that affect the health status of individuals. These are what the individual knows, what he believes, and how he behaves. These elements are the basic parts that make up a person’s survival and health. It is reported in the literature that the knowledge and attitudes that affect this behavior are as important as how a person behaves in order to stay healthy [15]. Based on the literature, a scale development study for COVID-19 was conducted based on the view that the individual would stay healthy by evaluating the three main elements together.

The conceptual framework consists of three main components. These are what the individual knows and believes and how he behaves. These three main structures also consist of sub-components within themselves. For this reason, structural models consisting of sub-dimensions were created for each of the three different conceptual frameworks. Accordingly, what individuals need to know about the disease is how the disease is transmitted and how to be protected. In other words, the knowledge dimension consists of disease, “contamination” and “protection” dimensions. From the perspective of health protection, it is mainly aimed at preventing individuals from contracting COVID-19. In order to achieve this, the individual must know how to protect himself from the disease and the basic transmission routes. The two basic substructures of the knowledge dimension are composed of the contagion and prevention knowledge dimensions.

The second important concept of the research is the individual’s attitude towards the disease. As it is known, individuals have various views and beliefs about their health and illness. There may be sufficient or insufficient information about these views and beliefs, as well as previous experiences and good manners of the environment. In other words, the individual has a certain attitude towards illness or health. Almost all of the attitudes formed in the face of information or uncertainty that feeds this attitude are prejudices formed without a rational basis. The sum of these prejudices forms the individual’s belief about health. In the literature, it is accepted that belief in health is the antecedent of the behavior of the individual. Although the relationship between attitude and behavior about health does not overlap exactly, it is seen that one of them supports the other to a large extent. The most well-known conceptual model of health belief is the structure consisting of perceived susceptibility, seriousness, benefit and barrier dimensions. A six-dimensional structure was created by adding readiness and motivation dimensions to this structure. In this study, it was desired to learn how the feelings and thoughts of individuals about the disease evolved into an attitude. In this context, the study aimed to measure the health belief, in other words, the attitude of individuals through the first four dimensions.

The third concept is health behavior. Health behavior appears in two ways. The first is to avoid the bad, and the second is to seek the good. How and at what level these behaviors are performed will show how effectively the individual applies healthy behavior. On the other hand, knowledge and attitude from previous conceptual models will also shape behavior and form a larger structural model. In this study, first of all, the presence and level of seeking and avoidance behaviors and the structure were tested, and then the final test was carried out with the structural model in which knowledge, attitude and behavior were evaluated together within a larger conceptual framework.

The aim of this study was to develop the COVID-19 knowledge, attitude and behavior scale and to evaluate its psychometric properties.

## 2. Materials and Methods

### 2.1. Study Design and Setting

This research consists of a methodological study for scale development in order to measure knowledge, attitudes and behaviors towards COVID-19. The research was carried out in two separate enterprises located in the Organized Industrial Zone in Manisa and Eskisehir in Turkey between August and November 2020.

### 2.2. Data Collection and Sampling

The research was performed with 543 employees (white and blue collar), including 349 employees (47.49%) in an enterprise with 735 employees in Manisa, and 194 employees (47.9%) in two enterprises with 405 employees in Eskisehir, in organized industrial zones. An attempt was made to reach all of the employees in the study, and to reach without sampling all employees aged 18 and over who volunteered to participate in the research and who were at work during the time the research data were collected. It was taken into consideration that the number of scale items used in the determination of the research group was 5–10 times higher [16].

### 2.3. Data Collection Tools

Within the context of this research, “Personal Information Form”, “Occupational Health and Safety Scale (OHSS)”, “Occupational Safety Climate Scale (OSCS)”, Health Literacy scales and COVID-19 Knowledge, Attitude, Behavior Scale Draft (CKABSD) were use data collection tools.

(a)Personal Information Form: Considering the purpose of the research, the information form consisted of 21 questions which included questions about age, gender, marital status, educational status, smoking, physical activity, health service use, general health and lifestyle related to health perception, as well as about working conditions such as daily working time, working style, professional experience, occupational health and safety, getting education and having a work accident [2,8,17,18].(b)COVID-19 Knowledge, Attitude and Behavior Scale (CKABS): The CKABS draft included a pool of 72 questions in total, with 27 for the knowledge level, 25 for the attitude level and 20 for the behavior level, which was created as a result of the literature review [2,6,14,19]. During the data collection phase, the entire item pool was questioned. It was attempted to create scale questions and answer options in a language that would appeal to all education groups. For this reason, three-point Likert-type scaling was used for the knowledge and attitude questions of the scale. Response options were formed as 1—disagree, 2—partly agree, and 3—totally agree. The response options given to the behavior dimension questions were structured to measure the state and frequency of the behavior. Accordingly, the response options were 1—never, 2—sometimes, 3—often and 4—always. In scoring the answers given to the questions, each sub-dimension and main dimension were scored separately, scaling out of 100. Score calculation formulas are given in the Appendix B Table A1. As a result of the psychometric analyses applied in line with the conceptual framework previously constructed, item reduction was performed; items with the highest model fit and items with a good level of item–dimension correlation among repetitive questions measuring the same concept were kept in the scale. Items with low item–dimension concordance, low or high correlation between items, and insufficient item discrimination were excluded from the scale. The scale dimensions and item numbers formed after the analysis are as follows: The knowledge dimension of the scale aims to measure the knowledge of individuals about the COVID-19 disease about contagion (KC) (3 items) and protection (KP) (7 items). The attitude dimension of the scale is based on the Health Belief Model (HBM). It consists of perceived susceptibility (ASus) (4 items), severity (ASe) (4 items), benefit (ABen) (2 items) and barrier (ABar) (3 items) sub-dimensions. Finally, the behavior dimension consists of the sub-dimensions of seeking-health behavior (BS) (6 items) and avoidance of illness (BA) (6 items).(c)Occupational Health and Safety Awareness Scale: It is a scale developed by Pehlivan to determine the level of occupational health and safety awareness. The 5-point Likert-type scale consists of 20 questions and as the score increases, the awareness of occupational health and safety increases [18].(d)Occupational Safety Climate Scale (OSCS): It is a 5-point Likert-type scale consisting of 14 questions which is adapted to Turkish by Turen et al., developed to determine the occupational safety climate in the construction industry [20].(e)Health Literacy Scale (HLS-32): The scale was developed by Okyay et al. It is a 5-point Likert-type scale with 32 questions [21]. The scale includes two health-related dimensions (treatment, disease prevention and health promotion) and four processes of obtaining information about health-related decision-making and practices (access, understanding, evaluation, and use/disuse). 0-25 points indicates insufficient health literacy, >25-33 points problematic/limited health literacy, >33-42 points indicate adequate health literacy and >42-50 points indicates excellent health literacy.

### 2.4. Content Validity and Piloting

An item pool was created by the research team based on the literature, aiming to measure all three basic constructs [4,6,22,23]. At this stage, expert opinion was sought in order to evaluate the relevance of the scale items to the subject, the necessity of the questions, and their simplicity, clarity, intelligibility, specificity and suitability for the target audience. Qualitative opinions were received from five experts regarding this item pool, and a scale consisting of 72 items in total was developed. Twenty-seven of these items measure the level of knowledge, 25 of them attitude and 20 of them behavior (Appendix A). Before the field application of the scale, it was updated by making a preliminary application to 40 people. The scale, which took its final form for the main application, was applied to the target group.

### 2.5. Psychometric Analysis of the Scale

#### 2.5.1. Construct Validity

At this stage, which includes scale validity and reliability analysis, explanatory and confirmatory approaches were applied together, and the final version of the COVID-19 knowledge–attitude–behavior scale was created. The scales measure different conceptual frameworks. For this reason, each scale (knowledge, attitude and behavior scales) was analyzed separately in the analysis process. Each of the scales was subjected to explanatory factor analysis according to its dimensions, which were formed as a whole and conceptually. Principal component analysis was applied in explanatory factor analysis, and varimax rotation was used for rotation [24,25].

Parallel to the exploratory factor analysis applied, the items containing the scale dimensions were separately analyzed in terms of Cronbach’s alpha value of internal consistency coefficient, Cronbach’s alpha value when the item was deleted, and item–total correlation coefficient. According to the results obtained, the items that did not show sufficient harmony with their own dimensions were reduced. After these stages, the final version of the scale was created [26,27].

#### 2.5.2. Parallel Scale Validity

It is expected that the sub-dimensions and the total score of the scale will have a high correlation with the scores obtained from the scales that are thought to measure similar concepts. Within the scope of Parallel Scale Validity, Occupational Health and Safety Awareness Scale, Occupational Safety Climate Scale (OSCS) and Turkish Health Literacy Scale (THLS-32) scales, which are explained in detail in the Data Collection Tools section, were used [18,20,21].

### 2.6. Statistical Analysis

Prior to calculating the psychometric parameters of the scale, the discrimination of the items was tested. Correlation coefficients for each of the 75 items in total with their own dimensions were examined. Items with a correlation coefficient below 0.5 were excluded from the dimension. In addition, a Mann–Whitney U test was applied in the lower and upper quartiles for item discrimination (Appendix B Table A2). The validity and reliability of the COVID-19 Knowledge Attitude Behavior Scale with 35 items was evaluated by psychometric analysis (Appendix A).

Descriptive analysis: The mean, median, standard deviation, min-max, skewness and kurtosis values are presented.

Reliability analysis: Reliability analysis is demonstrated by the “item analysis” and “internal consistency” approaches. In item analysis, corrected overlap correlation values were obtained according to the overlap between each question score and the total score, and the contribution of the questions to the scale was examined. Internal consistency was demonstrated by Cronbach’s alpha internal consistency coefficient. As an indicator of its invariance over time, the scale was reapplied to 50 randomly selected people after a two-week interval. The intraclass correlation coefficient (Intraclass Correlation Coefficient, ICC) of the scale dimensions and sub-dimensions was calculated between the repetition results and the first application results.

Validity analysis: In the validity analysis, the criterion validity and structural validity of the CKABSD were evaluated. Known groups in construct validity were evaluated with Explanatory (principal component analysis, Varimax cycle) and Confirmatory Factor Analysis (CFA) approaches. In CFA, Comparative Fit Index (CFI), Root Mean Square Error of Approximation (RMSEA), Standardized Root Mean Square Residual (S-RMR), chi-square/degree of freedom and Goodness of Fit Index (GFI) values were calculated. The values were minimum 0.90 for CFI and GFI, 0.08 maximum for RMSEA and 0.05 maximum S-RMR; chi-square/degree of freedom should be <3 [28,29]. Correlation analysis was performed with the Occupational Health and Safety Awareness Scale and the Occupational Safety Climate Scale for known group validity. In order to test the relationship of the scale between health behavior and knowledge and attitude, first of all, correlation coefficients between dimensions were examined, and then multivariate linear regression analysis was applied. By testing the relationship of knowledge and attitude to behavior with structural equation modeling, the compatibility of this relationship was also examined. Thus, conceptually, the relationship between knowledge and attitude and behavior is shown. SPSS 25.0 (IBM Inc., Armonk, NY, USA), Jasp 0.14 (University of Amsterdam) and Lisrel 9.1 (Scientific Software International, Inc., Chapel Hill, NC, USA) statistical package programs were used in the analysis.

### 2.7. Ethical Approval

Before starting the study, permission was obtained from the Ministry of Health and the Ethics Committee of Izmir Bozyaka Training and Research Hospital (2019/247). In addition, verbal consent to participate in the study was obtained from the participants.

## 3. Results

Of all the participants, 83.1% were male, the mean age was 37.4±8.0, 76.1% were married, 49.4% were high school graduates and 23.6% were university graduates. In addition, 13.1% of the participants reported that they had a chronic disease and 84.9% of them reported that their knowledge about COVID-19 was sufficient. The distribution characteristics of the scale items are presented in Appendix B Table A3. The knowledge part of the COVID-19 scale consists of 10 items and two dimensions. The Cronbach alpha value of the COVID-19 knowledge section is 0.86 in total, the contagion knowledge dimension is 0.71 and the prevention knowledge dimension is 0.84. When items are deleted in both sub-dimensions, there is no item that increases the Cronbach’s alpha value. It is seen that the item–total correlation adjusted for overlap in each dimension is not less than 0.4. The test–retest consistency ICC value for the knowledge dimension of the scale was 0.83, 0.71 for the contagion knowledge sub-dimension, and 0.87 for the prevention knowledge sub-dimension (Table 1).

The COVID-19 attitude section consists of four sub-dimensions and 13 items classified within the framework of SIM. Cronbach’s alpha values of perceived susceptibility, severity, benefit and disability sub-dimensions in the attitude section are 0.77, 0.75, 0.73, 0.59 and 0.70, respectively. Cronbach’s alpha value did not increase when the item was deleted for each sub-dimension, and the item–total correlations were above 0.4. The test–retest consistency ICC value for the total attitude scale was 0.93, and it was 0.91, 0.96, 0.88, and 0.87 for the perceived susceptibility, severity, benefit, and disability sub-dimensions, respectively (Table 2).

The COVID-19 behavior dimension consists of 12 items in total. Correct behavior seeking and avoidance sub-dimensions each consist of six items. The Cronbach’s alpha values of the total and its dimensions are 0.83, 0.80 and 0.74, respectively. It was determined that when the items of the scale were deleted, the Cronbach’s alpha value did not increase and the corrected item–total correlations were above 0.4. The test–retest consistency ICC value for the behavior dimension is 0.93, while it is 0.93 for the seeking sub-dimension and 0.79 for the avoidance sub-dimension (Table 3). The Cronbach’s alpha value for the scale total is 0.84.

The test–retest consistency ICC value for the sum of knowledge–attitude–behavior, which is the whole of the scale, is 0.89. Confirmatory factor analysis was applied for each part of the scale separately. Summary goodness of fit values are χ^2^/sd 2.29, 1.77 ve 2.57 for knowledge, attitude and behavior sections, respectively, indicating acceptable and good fit. RMSEA values of 0.047, 0.037 and 0.053, respectively, were sufficient; CFI values were 0.989, 0.984 and 0.976; GFI values were 0.973, 0.971 and 0.960. It is seen that both the comparative goodness of fit values and the goodness of fit values are at a good level (Figure 1).

Correlation of knowledge, attitude and behavior parts of the scale with the sum of occupational health and safety awareness scale was r = 0.317, r = −0.259 and r = 0.170, respectively (*p* < 0.001). It is seen that there is a significant correlation at the level of r = 0.313 between the Occupational Health and Safety awareness scale and the COVID-19 knowledge–attitude–behavior total score. The scale also shows a low but significant correlation with the health literacy scale (r = 0.118, *p* < 0.01) (Table 4).

When the correlation coefficients of the scales with each other are examined, it is seen that the majority of them show significant associations at low, medium and high levels. Significant association of behavior total score and all dimensions is observed (Table 5).

When the behavior total score was accepted as the dependent variable, it was determined that both variables had a significant relationship in the first model created with the knowledge and attitude total scores. The descriptive coefficient of the first model is R^2^= 0.838. This figure may be an indication that knowledge and attitude predict behavior at a very high level (Table 6).

In the second model, the sub-dimensions of knowledge and attitude were analyzed against the behavior total score. All of the sub-dimensions of knowledge and attitude were found to be significantly associated with the behavior total score. The descriptive coefficient of the model is R^2^ = 0.838. It was determined that all sub-dimensions predicted behavior independently of each other (Table 6).

According to the result of structural equation modeling created between the behavior total scale and knowledge and attitude scales, it was determined that both knowledge and attitude total scores were significantly related. When the summary goodness of fit values of the created model are examined, it is at an acceptable level of compliance considering the results as RMSEA = 0.078, GFI = 0.97, Stand.RMR = 0.057 and CFI = 0.92. In the model, both knowledge and attitude total scores of 0.20 each predict behavior at the standardized beta level. When the association and relations between the scale and the behavior dimension are evaluated as a whole, it can be determined that it is successful and distinctive in predicting behavior (Figure 2).

## 4. Discussion

COVID-19 is a serious threat to public health. Effective disease prevention will require the efforts of a significant portion of the world’s population, including social distancing and avoiding unnecessary interactions with others. No matter how rigorously these measures may be enforced, some people may not believe they will work or that they can act in ways that limit the threat. It is stated that an individual’s proficiency in improving his/her preventive behaviors largely depends on the risk of contracting a disease, and risk perception is very important in predicting these behaviors [4,30]. HBM, as it is known, is a model used to examine behaviors related to prevention or mitigation of disease. Individuals’ beliefs about illness are conceptualized as their perception of the severity of the threat, the perception of the benefits and barriers of action, and the individual’s perception of their intrinsic ability to act. In order to strengthen behaviors to prevent the spread of COVID-19, the severity of and susceptibility to the threat should be underlined, the barriers to action should be emphasized and individuals should be helped to fight the epidemic.

HBM is applied in many areas such as smoking cessation [31], colon [4], cervix [32] and breast [33] cancer awareness and screening, diabetic foot care [23], increasing flu vaccination rates [34], physical activity in the elderly [35] and rational use of antibiotics [36]. The wide application areas of HBM include infectious diseases such as Ebola, dengue fever, SARS, H5N1 and nosocomial infections [37]. Regarding the COVID-19 pandemic, which has emerged as a global problem, HBM can form a useful basis for measurement tools developed to determine the awareness levels of individuals on the subject, the attitudes they develop and the health-seeking and protection behaviors they display in this direction. The Cronbach α level was calculated as 0.83 for the knowledge scale, 0.77 for the attitude scale and 0.84 for the behavior scale. These values are above 0.80 for the knowledge and behavior scales and are very close for attitude, showing that the scale is highly reliable.

In a similar study conducted in Brazil on the same subject, the Cronbach α level was found to be 0.834 [38], and in a study conducted in Korea, it was found to be above 0.60 [39]; it was also determined between 0.785–0.712 in a scale adaptation study conducted in our country [40]. When compared with other studies, the levels obtained in our study are considered to be excellent in terms of reliability.

Behaviors towards protection from COVID-19 infection are closely related to socioeconomic level and are affected by regional, political and individual factors. In addition, unfounded news spread in different ways negatively affects the ability of individuals to develop desired attitudes and therefore their behavior in this regard [41]. In relation to COVID-19, knowledge, attitude and behavior studies have been carried out in various segments of society and in different countries based on HBM.

There are studies conducted on healthcare professionals and/or medical students [2,14,37,42,43,44] community-based studies [12,39,45,46,47,48,49,50,51,52,53], adolescents [54] and employees [55]. An original COVID-19 knowledge, attitude and behavior scale has not been developed in the literature; a scale was adapted from abroad [40]. There are very few studies on employees in the world, and no such study has been made in the literature. There is a need for a valid and reliable measurement tool in order to determine protection strategies for the employees who constitute the productive power of society.

In the regression analysis of the developed scale, when the total score of behavior is accepted as the dependent variable, both variables are significantly related in the first model using knowledge and attitude total scores and also in the second model using knowledge and attitude sub-dimensions. The descriptive coefficient of both models is R^2^=0.838. Accordingly, knowledge and attitude predicts behavior at a very high level. These calculated coefficients were higher than the descriptive coefficient calculated in a study conducted with adolescents in Iran (R^2^ = 0.46, *p* < 0.001) [54] and higher than the population-based study conducted in the same country (R^2^ = 0.509) [53]; thus, the model was evaluated as explanatory. In a scale adaptation study conducted in our country, the explanatory coefficient was calculated as 60.20% [40]. The proportion of those with sufficient knowledge of COVID-19 in knowledge in attitude and behavior studies was 48.3% in a study conducted in Bangladesh [45], 63% in Iran [51], 80.5% in Malaysia [12] and 88.0% in India [50]. According to the results of a review study including healthcare professionals, medical and non-medical students, and the general population, in which COVID-19 knowledge, attitude and behavior studies were evaluated, their knowledge levels were evaluated as sufficient in all six studies [2]. In the study of developing a COVID-19 knowledge and behavior scale in Iran, the level of knowledge showed the highest effect on the behavior of the participants, according to multivariate analysis [53]. It is seen that there is a sufficient relationship between the total score of the knowledge scale developed in this study and the scores of the contagion and prevention sub-dimensions, and that the knowledge level of the participants shows a positive correlation with the behavior both in total and in its sub-dimensions. Perceived susceptibility and perceived severity together can be defined as perceived threat and, according to HBM, increased threat perception can be attributed to health promotion behaviors. In our example, it can contribute to the implementation of behaviors aimed at preventing a pandemic. In our study, significant correlation coefficients were calculated with behavioral total scores (at the level of 0.487 and 0.426, respectively), and the result supports this argument. In addition, there are studies that do not have a relationship with perceived severity and susceptibility [47,48]. There are also studies in which it is effective in the first place among protection practices [55] and in the second place after the fatalistic understanding [49]. Perceived susceptibility was found to be effective in a study conducted in Singapore [56]. It has been reported to increase protective behaviors by 1.60 (95% CI: 1.06–2.39) times in Ethiopia [43]. In a study conducted in Korea [39], and in a study conducted on adolescents in Iran [54], it was reported that there was a negative correlation between protective behaviors and perceived susceptibility.

Perceived severity, the other component of perceived threat, was found to be unrelated to protective behaviors in a study conducted in Belgium [47], while it was evaluated as effective in the first place in a study conducted in ophthalmologists in Saudi Arabia [37]. In a study conducted in Korea [39] and in a study on adolescents in Iran [54], it was reported that there was a positive correlation between protective behaviors and perceived severity. In studies conducted in Singapore [46] and Ethiopia [55], perceived severity was found to be effective in determining protective behaviors, as in our study. Perceived benefit is another sub-component of attitude and its relationship with protective behaviors should be evaluated. The susceptibility of the model developed on the basis of HBM regarding COVID-19 in Singapore is 90.9% [46]. An HBM-based model created in Belgium, which explains 32.30% of the variance, shows that perceived benefit is also related to protective behaviors [47]. In a study conducted in China which reported that HBM can be used to assess individual differences in compliance with COVID-19 measures, perceived benefit was also associated with protective behaviors [48]. In a study in which it was stated that COVID-19 measures were insufficient in Ethiopia, perceived benefit was ranked above protective behaviors; it was found to be effective in the fourth place after perceived barrier, susceptibility and severity [55]. In a community-based study conducted in Iran based on HBM, in which fatalism was found to be effective in determining protective behaviors in the first place and women and urban residents were more compliant with the rules, perceived benefit had the least effect on protective behaviors [49]. Perceived benefit was in the second place following perceived severity in Saudi Arabian ophthalmologists [37]. In Ethiopia, it was found to be 0.35 effective (95% CI: 0.23–0.56), in line with self-efficacy, in healthcare workers [43]. Perceived benefit showed a positive correlation with protective behaviors ina population-based study in Korea [39] and in adolescents in Iran [54]. In our study, perceived benefit showed a positive correlation with total attitude and total scale scores; it is also effective in regression models in which protective behavior determinants are evaluated.

There are many studies which have found a relationship between perceived barriers and protective behaviors [43,46,47,48,49,50,54,55]. In a model developed in Singapore, the sensitivity for the perceived barrier was 81.5% [46]. In studies in China and Belgium, where the perceived susceptibility [47,48] and perceived severity [47] were unrelated, the perceived barrier was associated with protective behaviors. In a study conducted in Ethiopia, it ranked first in terms of impact on protection practices [55]; in Iran, it was ranked third after fatalism and perceived susceptibility [49]. In a study conducted in India, those with low perceived barrier were 6.31 (95% CI: 2.78–14.34) times [50] more compliant; in Ethiopia, they were 3.36 (95% CI: 2.23–5.10) times were more compliant with the measures. Perceived disability negatively affects individuals’ adaptation to protective behaviors in HBM [49,54]. In our study, it showed a positive correlation between the perceived barrier, attitude total and total scale. It is also effective in regression models in which protective behavior determinants are evaluated.

The rate of desired behavior in COVID-19 knowledge, attitude and behavior studies was reported to be 31.2% in one study in Ethiopia [55], 44.1% in another [43], 55.1% in Bangladesh [45], 78% in Iran [51], 80.5% in Malaysia [12] and 93.8% in India [50]. The behavior dimension consists of search and avoidance sub-dimensions in the developed scale. It is stated that seeking-avoidance behavior is an important indicator of quality of life, with the individual’s assessment of his own health status [57]. In the literature, the concept of seeking–avoidance behavior was encountered until the quality of self-care of cancer patients improved [57]. HIV-positive patients were informed about the follow-up process [58], and patients who would be operated on under spinal anesthesia were informed about their own care [59]. Problems of accessing the right information sources about health and communication inequality are considered among the social determinants of health and are associated with health outcomes by influencing the behavior of individuals. The seek and avoidance sub-dimensions of the scale evaluate the practices of prevention of epidemics and behaviors that are part of daily life in normal periods, but which should be avoided during pandemic periods. There is a significant positive correlation between the behavior sub-dimensions and all sub-dimensions, and between each separate and the total scale. When the CFA results applied for knowledge, attitude and behavior scales are examined, it is seen that the summary goodness of fit values are within acceptable limits. In addition, it was determined that the associations of the sub-dimensions of each scale with the items were sufficient and the error variances were at an acceptable level.

## 5. Conclusions

It was determined that the internal consistency of the COVID-19 knowledge, attitude and behavior scale carried out on the employees was high and consistent, and the validity findings were sufficient. The scale is recommended as an applicable tool to measure COVID-19 knowledge, attitude and behavior both in working life and in the general population. It is recommended that the COVID-19 information, attitude and behavior scale be applied in community-based studies in different populations. In addition, considering that the level of information, attitude and behavior of COVID-19 has reached a certain level in society during the pandemic, it is recommended to create a short form of the scale. In addition, it is recommended that the scale be used as an independent variable in terms of the effect of knowledge, attitude and behavior level in studies where there are dependent variables such as attitude towards the COVID-19 vaccine. Finally, in this study, socio-demographic characteristics were found to be associated with COVID-19 information, attitudes and behaviors. Therefore, in addition to socio-demographic factors, other determinants of KAP should be studied extensively.

### Limitations

A limitation of the study is that HBM can be associated with underlying demographic characteristics rather than being directly associated with structures such as social norms and presuppositions. HBM has not previously been used for any pandemic.

## Figures and Tables

**Figure 1 vaccines-11-00317-f001:**
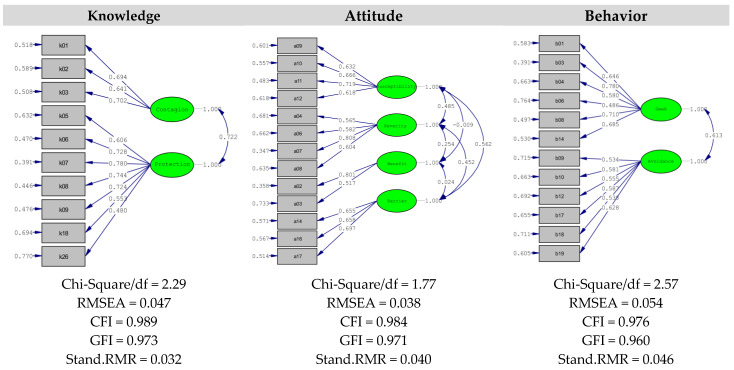
Confirmatory Factor Analysis Results. RMSEA: The Root Mean Square Error of Approximation, CFI: The Comparative Fit Index, GFI: Goodness of Fit Index, Stand.RMR: Standardized) Root Mean Square Residual.

**Figure 2 vaccines-11-00317-f002:**
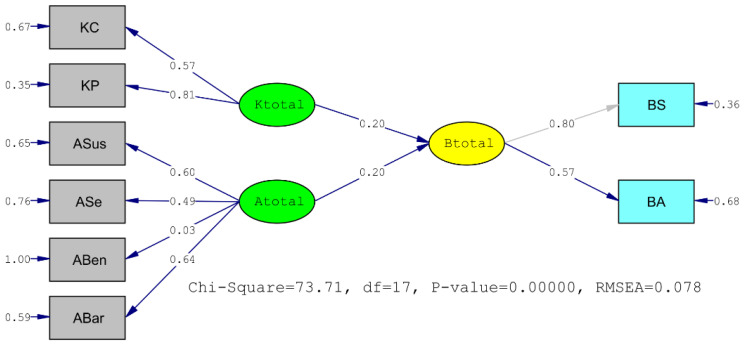
The Relationship Between Knowledge, Attitude and Behavior Scales, Structural Equation Modeling Analysis. RMSEA: The Root Mean Square Error of Approximation.

**Table 1 vaccines-11-00317-t001:** Knowledge dimension reliability findings.

Knowledge Dimension	Adjusted Item–Total Correlation	Cronbach’s Alpha When Item Is Deleted	Cronbach’s Alpha Dimension (ICC) ^#^	Cronbach’s Alpha Total (ICC) ^#^
Knowledge–Contagion Dimension				
KC1-COVID-19 disease is transmitted by droplets in the coughs of patients.	0.55	0.59	0.71 (0.71)	0.86 (0.83)
KC2-COVID-19 is more severe in the elderly and those with chronic diseases.	0.50	0.67		
KC3-The most common symptoms of COVID-19 disease are fever, cough and respiratory distress.	0.54	0.60		
Knowledge–Protection Dimension				
KP1-Masks should be worn to protect against COVID-19 disease.	0.54	0.83	0.84 (0.87)	
KP2-To prevent the virus, hands should be washed with soap and water.	0.69	0.81		
KP3-In the absence of water and soap, alcohol-containing disinfectant or cologne should be used.	0.72	0.80		
KP4-Contact of contaminated hands with eyes, mouth and nose may cause disease.	0.67	0.81		
KP5-Everyone who comes together to protect from the disease must wear a mask.	0.67	0.81		
KP6-In order to protect from disease in workplaces, the rules of distance, hygiene and wearing masks should be observed among employees.	0.50	0.84		
PK7-The mask should be changed when it becomes damp or soiled.	0.42	0.84		

^#^ Intraclass Correlation Coefficient (ICC).

**Table 2 vaccines-11-00317-t002:** Attitude dimension reliability findings.

Attitude Dimension	Adjusted Item–Total Correlation	Cronbach’s Alpha When Item Is Deleted	Cronbach’s Alpha Dimension (ICC) ^#^	Cronbach’s Alpha Total (ICC) ^#^
Attitude–Susceptibility Dimension				
ASus1-I do not believe that COVID-19 disease is transmitted by the scattering of droplets in the inhaled air.	0.52	0.71	0.75 (0.91)	0.77 (0.93)
ASus2-I do not believe that COVID-19 is transmitted by those who have had the disease asymptomatically.	0.56	0.68		
ASus3-I do not believe that wearing a mask protects people from COVID-19 disease.	0.60	0.67		
ASus4-I do not think it is necessary to wash hands with soap and water to be protected from COVID-19 disease.	0.51	0.71		
Attitude–Severity Dimension				
Ase1-COVID-19 cannot be easily transmitted to me.	0.46	0.70	0.73 (0.96)	
ASe2-COVID-19 is not a deadly disease according to me.	C	0.69		
ASe3-Even if I get COVID-19, I can easily get over it.	0.66	0.58		
ASe4-I think my body resistance to COVID-19 disease is quite high.	0.49	0.69		
**Attitude–Benefit Dimension**				
ABen1-Wearing gloves protects me from the disease.	0.41	-	0.59 (0.88)	
ABen2-I won’t get sick if I wear a mask outside.	0.41	-		
**Attitude–Barrier Dimension**				
ABar1-Social distancing is not important according to me in COVID-19.	0.49	0.64	0.70 (0.87)	
ABar2-Only the elderly and chronic patients should be protected in the community for COVID-19 disease.	0.51	0.62		
ABar3-It will be sufficient if only the sick people wear a mask.	0.55	0.56		

^#^ Intraclass Correlation Coefficient (ICC).

**Table 3 vaccines-11-00317-t003:** Behavior dimension reliability findings.

Behaviour Dimension	Adjusted Item–Total Correlation	Cronbach’s Alpha When Item Is Deleted	Cronbach’s Alpha Dimension (ICC) ^#^	Cronbach’s Alpha Total (ICC) ^#^
Behavior–Seek Dimension				
BS1-I try to cover my mouth when I cough and sneeze.	0.55	0.77	0.80 (0.93)	0.84 (0.93)
BS2-To protect myself from COVID-19 infection, I wear a mask when I go out.	0.68	0.74		
BS3-I wear the mask to cover my mouth and nose.	0.51	0.78		
BS4-In the absence of water and soap, I use alcohol-containing hand sanitizer or cologne.	0.43	0.80		
BS5-I wear a mask on public transport.	0.62	0.76		
BS6-When the disease is common, when I come home from outside, I first wash my hands with soap and water.	0.59	0.76		
Behavior–Avoidance Dimension				
BA1-I take a break from visiting family and friends when illness is common.	0.43	0.71	0.74 (0.79)	
BA2-I act in accordance with social distance in the working and resting environment.	0.52	0.69		
BA3-I act in accordance with social distance in public transport vehicles.	0.46	0.71		
BA4-When illness is common, I try not to be in enclosed spaces where other people are present.	0.47	0.70		
BA5-I pay attention to how other people behave outside.	0.47	0.71		
BA6-I pay attention to the social distance between me and other people when the disease is common.	0.51	0.70		

^#^ Intraclass Correlation Coefficient (ICC).

**Table 4 vaccines-11-00317-t004:** Correlation between Occupational Health and Safety Awareness Scale and THL Scale and CKABS.

CKABS and Sub-Dimensions	Occupational Health and Safety Awareness Scale	Treatment Services	Protection Development	THLS Total
Contagion knowledge	0.178 **	0.057	0.068	0.069
Protection knowledge	0.323 **	0.116 **	0.066	0.080
Knowledge Total Score	0.317 **	0.104 *	0.088 *	0.093 *
Susceptibility	0.290 **	0.118 **	0.096 *	0.104 *
Severity	0.209 **	0.018	−0.064	−0.025
Benefit	−0.036	0.049	0.067	0.048
Barrier	0.232 **	0.163 **	0.072	0.120 **
Attitude Total Score	0.259 **	0.118 **	0.060	0.077
Seek behavior	0.170 **	0.122 **	0.058	0.106 *
Avoidance behavior	0.146 **	0.128 **	0.111 *	0.129 **
Behavior Total Score	0.170 **	0.132 **	0.102 *	0.131 **
CKABS Total Score	0.313 **	0.147 **	0.097 *	0.118 **

* *p* < 0.05, ** *p* < 0.01.

**Table 5 vaccines-11-00317-t005:** Correlation Between CKABS Sub-Dimensions and Total Scores.

CKABS and Sub-Dimensions	KC	KP	KT	Asus	ASe	ABen	ABar	AT	BS	BA	BT
Contagion knowledge											
Protection knowledge	0.568 **										
Knowledge Total Score	0.810 **	0.943 **									
Attitude susceptibility	0.137 **	0.164 **	0.172 **								
Attitude severity	0.195 **	0.180 **	0.207 **	0.369 **							
Attitude benefit	0.114 **	0.210 **	0.196 **	−0.023	−0.186 **						
Attitude barrier	0.236 **	0.326 **	0.328 **	0.405 **	0.339 **	−0.023					
Attitude Total Score	0.279 **	0.366 **	0.374 **	0.728 **	0.576 **	0.422 **	0.655 **				
Behavior seek	0.134 **	0.217 **	0.208 **	0.122 **	0.116 **	0.103 **	0.206 **	0.225 **			
Behavior avoidance	0.126 **	0.163 **	0.167 **	0.110 *	0.094 *	0.122 **	0.105 **	0.186 **	0.499 **		
Behavior Total Score	0.150 **	0.219 **	0.217 **	0.134 **	0.121 **	0.130 **	0.179 **	0.237 **	0.865 **	0.866 **	
CKABS Total Score	0.593 **	0.728 **	0.759 **	0.487 **	0.426 **	0.351 **	0.547 **	0.758 **	0.572 **	0.533 **	0.638 **

* *p* < 0.05, ** *p* < 0.01. KC: COVID-19 disease about contagion, KP: Knowledge protection, Asus: The attitude perceived susceptibility, ASe: The attitude severity, ABen: The attitude benefit, ABar: The attitude barrier, BS: The Behavior seeking-health behavior, BA: The attitude avoidance of illness.

**Table 6 vaccines-11-00317-t006:** The Relationship Between Behavior Total Score and Knowledge and Attitude, Linear Regression Analysis Models.

CKABS Dimensions and Sub-Dimensions		Beta	Standard Error	Standardize Beta	t	*p* Value	Tolerance	VIF
**Model 1** **R^2^ = 0.838**	Constant	20.508	1.215		16.877	0.000		
	Kowledge total score	0.376	0.013	0.553	29.568	0.000	0.860	1.162
	Attitude total score	0.387	0.013	0.551	29.489	0.000	0.860	1.162
**Model 2** **R^2^ = 0.838**	Constant	20.358	1.247		16.321	0.000		
	Contagion knowledge	0.104	0.011	0.207	9.716	0.000	0.667	1.499
	Protection knowledge	0.271	0.015	0.406	18.260	0.000	0.612	1.635
	Attitude Susceptibility	0.092	0.008	0.238	12.059	0.000	0.774	1.292
	Attitude severity	0.096	0.009	0.210	10.605	0.000	0.772	1.296
	Attitude benefit	0.100	0.006	0.292	15.951	0.000	0.902	1.109
	Attitude barrier	0.102	0.010	0.205	10.109	0.000	0.735	1.360

VIF: Variance Inflation Factor.

## Data Availability

Data is contained within the article and Appendix A.

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
