# Peer review of "Validity and Reliability of the COVID-19 Knowledge, Attitude and Behavior Scale"

_vaccines, 2023, doi:10.3390/vaccines11020317_

Round 1

Reviewer 1 Report

Dear authors,

I have now completed the review of the manuscript titled "Validity and Reliability of the Covid-19 Knowledge, Attitude 2 and Behavior Scale."

In the present study, the authors developed a scale that will evaluate the knowledge, attitudes and behaviors of employees about Covid-19. 

The manuscript is interesting and, in general, fair written.

I have some suggestions to further improve the quality of the manuscript.

1. The background section introduced some relevant articles. Please explain the results or summarize with effect sizes. 

2. I suggest authors clarify how other researchers can obtain the original data.

3. Authors used varimax rotation. I wonder this process is most proper for the current dataset. Could you suggest data statistics or references?

4. What is the future scope of the proposed research, authors have described the limitations in a good way, I suggest that these can be the future scope of the work.

Author Response

Dear referee, all suggestions were accepted and necessary corrections were made.

Reviewer 2 Report

The article presents the development and validation of the scale. It fills an existing gap and could be a valuable instrument for researchers. The statistical techniques used are adequate and sophisticated. However, the manuscript needs a thorough revision.

 1.       Introduction. Line 51 states, “the attitude score is 41-99”. It would be necessary to indicate the total scale to which it refers, for example, on a scale of 1 to 200 or 1 to 100.

2.       The article is disorganized—section 2.4. Covid-19 Knowledge-Attitude-Behavior Scale Development and Analysis Steps is included in the material and methods but should be moved to the introduction.

3.       Data Collection and Sampling. The authors should explain the procedure for selecting participants. It was a random sampling, a convenience sampling.

4.       Line 105 states that Score calculation formulas are given in the appendix. However, there is no appendix.

 5.       The entire questionnaire used translated into English should be included as an appendix.

6.       The text of the article should include the final scale and the instructions for its correction.

7.       In line 209, it is indicated, “Descriptive analyzes: The mean, median, standard deviation, skewness floor and  ceiling percentages of individual items and total index score are presented.” However, this information is nowhere to be found in the article.

8.       In the design phase, prior to the calculation of the psychometric parameters of the scale, it is necessary to assess those items that are susceptible to elimination. It would be necessary to measure the discriminative capacity of the items. This should be done in two ways.

·         Authors should correlate each item of the scale with the total score of the scale. Those items with a low correlation should be candidates for elimination from the scale.

·         The sample should be divided into quartiles of the total scale score. Comparisons should then be made between the first and fourth quartiles of the scale with each of the items using the Mann-Whitney U test. Items that were not significant could be eliminated.

9.       Authors should present a table with Pearson correlations and another with the U Mann Whitney test, indicating, e.g., with an asterisk, those items susceptible to elimination.

10.   The authors have calculated parameters such as Cronbach’s alpha for the subscales. They should provide the total Cronbach’s alpha of the scale.

11.   References. The numbering of the bibliography is repeated. This should be revised

12.   Figure 1 is difficult to read. Try to improve the resolution.

13.   The section on material and methods is very long and should be synthesized. For example, when it mentions the scales used for Parallel Scale Validity, it would be sufficient to mention them and include the bibliography.

Author Response

(The authors gave the same response as above.)
